# The Application of an Allogenic Bone Screw for Stabilization of a Modified Chevron Osteotomy: A Prospective Analysis

**DOI:** 10.3390/jcm11051384

**Published:** 2022-03-03

**Authors:** Thorsten Huber, Stefan G. Hofstätter, Rainer Fiala, Florian Hartenbach, Robert Breuer, Björn Rath

**Affiliations:** 1Department of Orthopedics and Trauma Surgery, Klinikum Wels-Grieskirchen, 4600 Wels, Austria; florian.hartenbach@gmail.com (F.H.); robert.breuer@meduniwien.ac.at (R.B.); bjoern.rath@klinikum-wegr.at (B.R.); 2Independent Researcher, 4600 Wels, Austria; stefanghofstaetter@gmail.com; 3Department of Orthopedics, Klinik Donaustadt, 1220 Vienna, Austria; rwf3a@virginia.edu

**Keywords:** allogenic bone screw, osteointegration, distal first metatarsal osteotomy

## Abstract

Background: Distal first metatarsal osteotomies are commonly performed operative procedures for hallux valgus deformity, and usually involve fixation with a metal screw. However, various bioabsorbable osteosynthesis materials have been in use for a number of years. One recent innovation is the Shark Screw^®^, a human cortical bone allograft. This study aimed to evaluate the efficacy and safety of this allogeneic screw in the stabilization of Reversed L-Shaped osteotomy, a modified Chevron osteotomy. Methods: In a prospective study, 15 patients underwent a Reversed L-Shaped osteotomy stabilized with the allogenic bone screw Radiological data on osteointegration of the screw and correction of the intermetatarsal angle were recorded. Furthermore, each follow-up examination included the collection of clinical data, the American Orthopedic Foot and Ankle Society (AOFAS) score, evaluation of pain level, and patient’s overall satisfaction. Results: Full osseous fusion of the osteotomy was seen in all patients. The bone screws were radiographically integrated after approximately 6.5 (±2.6) months. Neither nonunion nor failure occurred in any of our cases. Furthermore, we did not find any potential graft reaction. The AOFAS score improved significantly from 51.6 (±15.2) points to 90.9 (±10.3) (*p* < 0.001). The preoperative hallux valgus angle and intermetatarsal angle decreased significantly from 24.8 (±4.9) degrees to 7.2 (±4.4) degrees (*p* < 0.001) and 12.6 (±3.2) degrees to 4.8 (±1.3) degrees (*p* < 0.001), respectively. Conclusions: With this study, we demonstrated the efficiency of the allogenic bone screw (Shark Screw^®^) in regard to clinical and radiological short-term outcomes.

## 1. Introduction

The painful hallux valgus deformity is a common orthopedic problem that occurs mainly in women. Hallux valgus surgery is frequently performed with different surgical techniques and treatment algorithms. A common technique for treating a moderate hallux valgus (HV) deformity is the Chevron osteotomy, a distal first metatarsal osteotomy. There are many modifications of the technique described by Austin et al. in 1981 [1,2,3,4].

A well-known modification of the previously described technique is the Reversed L-Shape (ReveL) osteotomy by Espinosa et al. [5]. This technique combines the advantages of the previously mentioned Chevron osteotomy and the scarf osteotomy, and is known as a diaphyseal metatarsal osteotomy [6].

Metallic screws are commonly used for osteotomy stabilization, providing the advantage of high stiffness, good compression capabilities, and corrosion-resistance in titanium alloys. However, as they commonly remain within the body as foreign material, they may cause possible complications such as allergic reactions, soft tissue irritation, and subsequent pain. In some cases, a surgical procedure for metal removal is necessary. The economic impact, in particular the necessary sick leave, as well as the cost of the procedure itself should also be taken into account [7,8,9,10].

Research is being conducted constantly to develop new materials and alternative stabilization options. Bioabsorbable osteosynthesis materials such as ceramic-based synthetic bone substitutes, mostly based on hydroxyapatite (HA) and tricalcium phosphate, are used as alternative osteosynthesis in maxillofacial surgery with good results. Although they are softer than cortical bone, they are partially effective [11,12].

The role of osteoconductive and osteoinductive materials is especially relevant in orthopedic surgery, for example, in the treatment of pseudarthrosis. It remains elusive as to whether artificial bone grafts are capable of weightbearing. Thus, weightbearing is necessary in early functional therapy and is a key principle of modern osteosynthesis in extremities.

However, an autologous bone graft is the gold standard material for bone regeneration in terms of osteoconduction and osteoinduction, but the problem of limited availability and donor site morbidity need to be taken into account. For this reason, allogenic bone material is often used. Allogenous bone is available in various shapes and sizes, and if necessary, it is coated with antibiotics. However, since the material has a low osteoinductive property, it may lead to inferior healing compared to autogenous bone [11].

The Shark Screw^®^ transplant (surgebright GmbH, Lichtenberg, Austria) is a human cortical bone allograft. It is a new osteosynthesis device that allows fracture-, osteotomy- and arthrodesis-fixation and it acts as an alternative to metal or bioabsorbable devices. The screw is designed as a set screw and is integrated into the recipients’ bone by a continuous bone remodeling process. A full conversion into autologous bone has been described [13].

This study aimed to critically evaluate the efficacy and safety of the allogeneic screw-in the stabilization of ReveL osteotomy. Our primary outcome parameter was the osteointegration of the screw into the patient’s bone.

## 2. Materials and Methods

This study was conducted following the Declaration of Helsinki and the Guidelines for Good Clinical Practice. Ethical approval was received from the Ethics Committee of Upper Austria (Vote-No: 1032/2018). The trial was registered at clinicaltrials.gov (NCT03884907).

The study used a prospective study design.

### 2.1. Patients

Fifteen patients/feet (12 female, 3 male) with an average age of 56.3 (±12.5) years at the time of surgery were included. In all patients, HV surgery with ReveL osteotomy was performed between August 2018 and March 2019. The osteotomy was fixed with an allogeneic bone screw (Shark Screw^®^). Additional procedures were performed in 5 cases (4 Weil osteotomies, 1 Akin osteotomy). The mean follow-up time was 17 months. One patient was lost to follow-up after 2 months due to personal reasons. Written informed consent was obtained from the patients (Table 1).

The indications for HV surgery followed the algorithm by Robinson and Limbers [14], including symptomatic HV deformity, IMA < 20 degrees, HVA < 60 degrees, no grade 3 and 4 cartilage damage of the metatarsophalangeal joint, and no hypermobility or instability or arthritis in the first tarsometatarsal joint (Table 2).

### 2.2. Surgical Technique and Follow-Up Treatment

The operation was performed according to a standard surgical protocol. A tourniquet was used during the surgery. The lateral release was performed before the osteotomy, either with an intermetatarsal skin incision or in a transarticular manner.

A longitudinal skin incision medial to the MTP I joint was undertaken. The cutaneous nerve was protected and the first step was to incise extra-articular to the interdigital space I/II. The capsule was opened longitudinally under visual control, and the successful lateral release was confirmed by achieving 20 degrees in the varus stress test. The medial pseudoexostosis was removed. A K-wire was inserted transversely in the center of the metatarsal head from medial to lateral and drilled in the direction of the third or fourth metatarsal head. The Reversed L-Shaped osteotomy was performed at an angle of about 60 degrees with a long plantar limb (Figure 1).

This provides a good bearing surface, greater correction potential, and the blood vessels supplying the head were protected. A correct “distal metatarsal articular angle” (DMAA) was ensured in this situation, and if necessary, was modified by additional dorsal osteotomy on the proximal fragment of the first metatarsal. The head fragment was then shifted laterally by up to 2/3 of the bone width so the IMA was reduced. The osteotomy was stabilized temporarily with a 1.2 mm K-wire from dorsomedial to plantolateral. Then, a 1.6 mm K-wire for the Shark Screw^®^ was inserted centrally from dorsal proximal to plantar distal; afterwards it was changed into a 1.2 mm K-wire and it was overdrilled. There are core drills of different thicknesses for different Shark Screw^®^ variants. In our case, only 4.0 mm diameter screws and a 3.25 mm core drill were used. Next, a thread was cut and the bone screw was screwed into the head fragment under counter pressure. The K-wires were then removed.

The protruding part of the screw, as well as the protruding bone on the medial side of the proximal metatarsal fragment were removed with the oscillating saw (Figure 2). Medially, the capsule was shirred and layered wound closure was performed. Postoperatively, an Elastoplast^®^ bandage was applied for additional stabilization and redressed for 6 weeks with weekly changes. Full weightbearing was allowed in a HV shoe from the first postoperative day for 6 weeks.

### 2.3. Clinical Examination, X-rays, and Scores

Standardized weightbearing AP radiographs of the foot were obtained for the measurement of the intermetatarsal angles (IMA) and hallux valgus angles (HVA), preoperatively and at each follow-up examination. The absence of radiolucent lines was used as a parameter to determine the consolidation of the transplant. A sclerotic rim around the transplant at the final follow-up would have been deemed as an absence of bony consolidation.

Examinations were performed preoperatively, on the first day postoperatively, and after 6 weeks, 6 months and 12 months postoperatively.

The surgeons were asked to assess the intraoperative bone quality as well as the primary stability of the osteosynthesis using a four-part scale consisting of very good, good, moderate, and poor.

Data on pre- and postoperative pain (numeric rating scale NRS 0–10) [15], and the American Orthopedic Foot and Ankle Society (AOFAS) score were collected at each follow-up visit [16].

Additionally, patient satisfaction (NRS 0–10) was analyzed at the last follow-up examination. The type and frequency of postoperative complications and soft tissue irritation were documented, in particular, swelling and wound healing disorders (graded on a 4-point scale: 0–3 none/minimal/moderate/strong, respectively). Dislocation or implant failure, nonunion, and the incidence of revision surgery were sought as well.

### 2.4. Statistics

Descriptive data are presented as mean and standard deviation (SD) for metric values as median, first and third quartiles for ordinal variables. Standard distribution testing was performed using the Kolmogorov–Smirnov test, and equality of variance using the Levene test. The difference in angle, AOFAS score, and level of pain pre- and postoperative, as well as during 12 months of follow-up were evaluated using the paired samples T-test for normally distributed values. All *p* values were 2-sided, with a significance level set at *p* ≤ 0.05. Statistical analysis was performed using the SPSS 24 (IBM, New York, NY, USA) and Microsoft Excel.

## 3. Results

The primary stability of the osteosynthesis was rated as very good in all cases. The quality of the patients’ metatarsal bone was classified as “very good” in 13 cases and as “good” in two cases. Zero cases were ranked as moderate or poor.

Full osseous fusion of the osteotomy was seen in all patients (n = 15). The screws were radiographically integrated after a mean time of 6.5 (±2.6) months. Radiolucent lines of more than 2 mm width or cystic lucencies around the transplant were detected in 5 cases after 2 and 6 weeks, but all of them disappeared, at latest, after 6 months postoperatively. There were no cases with a sclerotic rim around the transplant, which would indicate a failed bony healing. Neither nonunion nor failure occurred in any of our cases (Figure 3).

The AOFAS score improved significantly from 51.6 points preoperatively to 90.9 points (*p* < 0.001) at the final follow-up. The preoperative HVA and IMA decreased significantly postoperative (*p* < 0.001). In summary, physiological angles (HVA < 15 degrees and IMA < 9 degrees) were achieved. The pain value according to NRS decreased to a pain level of 1.0 (±1.4) points, and 50% of the patients were free of pain (NRS = 0) (Table 3).

We found delayed wound healing up to 6 weeks with prolonged swelling in three patients. Two of these patients were long-time smokers, who achieved complete wound healing at the 6-week follow-up visit. In the case of the third patient, the final wound closure was achieved two weeks later.

We did not find any potential graft reaction or allosensitization.

## 4. Discussion

The use of allogenic bone material in various fields of medicine is already widespread and the indications for it are constantly expanding. In orthopedic surgery in particular, but also in oral and maxillofacial surgery, allogenic bone is used in a multitude of variants [17,18,19].

This study demonstrates that the allogenic cortical bone screw, the Shark Screw^®^ is an efficient and safe substitute for a metal screw for ReveL osteotomy. A radiological bony fusion was confirmed in all of the patients. Radiological follow-up demonstrated the conversion of allogenic into autologous bone within the patients’ metatarsal.

Pastl et al. have already described the effective remodeling process of the Shark Screw^®^ in a wide variety of operations in hand and foot surgery [13]. As already mentioned in their study, we could not find potential graft reaction or allosensitization in any of our subjects as well.

The primary stability of the osteosynthesis in the course of our operations was described as very good by all surgeons. However, it must be taken into account that HV surgery with ReveL osteotomy, with a long plantar limb, has inherently high primary stability due to the surgical technique. In the past, some studies have even reported that there was no significant slippage of the osteotomy even without additional screw stabilization [20,21]. In some centers, only a K-wire is used as an alternative to the 3.0 mm head compression screw (HCS), which is routinely used in our clinic [2,3,22]. Thus, only limited information can be given about the additional stability achieved by the bone screw.

One advantage of a metal-free, osteointegrating fixation is the absence of hardware removal, which accounts for one of the most commonly performed surgeries in orthopedics [23,24,25]. According to the current literature, metal screws are routinely removed in up to 65% of patients. Material removal rates in forefoot surgery are about 7% of this group [8]. The complication rate after material removal, in general, is reported to be up to 20% in the literature [8,24]. The most common complications observed are infection, refracture, soft tissue and/or nerve damage, delayed and impaired wound healing, as well as postoperative bleeding and swelling [24]. Apart from the possible complications mentioned above, the socio-economic impact must be taken into consideration. Hardware removal is an expensive affair for both hospitals and health care resources [7,8,9,25].

Nevertheless, it should be mentioned that there are additional costs for the Shark Screw^®^ compared to a conventional HCS. The bone screw is 10–12 times more expensive, depending on the thickness of the screw. Considering the probability of material removal in HV operations, which has decreased since the use of the HCS, the use of the bone screw should be questioned in regard to the financial aspects.

However, it is important to consider that metal screws have high stiffness, and depending on the screw design, a compression effect is possible [8]. The Shark Screw^®^ is unable to produce a compression effect due to its shape, its continuous thread, and the lack of a screw head. The bone screw therefore can only be used as an adjusting screw. The postoperative regime did not differ between ReveL osteotomy with metal screws compared to the Shark Screw^®^ in our clinical practice. In other operations, the higher stiffness of metal screws is beneficial for earlier weightbearing. This reduces postoperative thromboembolic events and reduces the duration of the patient’s sick leave [26,27,28].

Compared to biodegradable osteosynthesis devices, the allogeneic bone screw is continuously remodeled into the body’s bone without producing degradation products. This comes from the interaction of osteocytes, osteoclasts, and osteoblasts [29]. The degradation of bioabsorbable polymers, in contrast, leads to a local inflammatory process [30,31]. This inflammatory reaction and the degradable phase leads to a weakening of the bone. Complications related to biodegradable osteosynthesis devices range from local pain, synovitis, and foreign material reaction to osteolysis [30,31,32,33].

The good ingrowth and remodeling behaviour of allogenic bone has already been described in other studies. Campana et al. stated that despite extensive research in the field of bone grafts, human bone replacement remains the gold standard. Alternatives lack one or more parts of Giannoudis’ diamond theory—osteogenic cells, vascularization, mechanical stability, growth factors, or an osteoconductive scaffold [11,34].

Allogenic bone material is inferior to autologous bone in terms of osteoinductivity. The harvesting of autologous bone, with its limited availability and the additional possible complications, especially wound infection, bone fracture, and pain is not necessary when using allogenic bone such as the Shark Screw^®^. However, in our opinion, the decision should be made according to the area of application and requires a thorough risk–benefit assessment.

According to the recommendation of the manufacturer, the leg should be unloaded for at least 2 weeks after an HV operation with the cortical bone screw. The company cites the protection of the sprouting stem cells with their “dendrites”, which could be injured and lose their differentiation potential. As a further consequence, the screw may also break. In our study, however, we did not observe any screw breakage and observed good remodeling behaviour despite allowing full weightbearing. As mentioned above, this may also be related to the high primary stability of the surgical technique.

In consideration of our secondary analysis points, this study showed significant improvements in the HVA, IMA, the reduction in the postoperative pain level, and the AOFAS score (*p* < 0.001). The Chevron osteotomy and modified surgical techniques are widely used. They are characterized by high patient satisfaction, good functionality, and low complication rates. We were able to reduce the IMA from 12.6 degrees to 4.8 degrees, which corresponds to comparable values in the literature [2,3,5,35,36]. The AOFAS score improved from 51.6 to 90.9 points. Comparable studies showed similar improvements [3,35,37].

### Strengths and Limitations of This Study

The strengths of this study are the prospective design, the homogenous patient characteristics, and the strict inclusion and exclusion criteria. The limitations of this study are the small number of participants and the midterm follow-up period. The reason is that it is a pilot study and the implant is new and expensive. Another limiting factor is the lack of a control group. By these reasons, the results should be interpreted with caution and further analyses are required. 

## 5. Conclusions

This study aimed to evaluate the radiographic osteointegration and efficacy of an allogenic bone screw (Shark Screw^®^) in the stabilization of the osteotomy in HV surgery. Given its high fusion rate and low rate of complication, the allogeneic bone screw could be an effective alternative to conventionally utilized fixation systems.

This study showed very good clinical and radiological outcomes. Patient satisfaction, especially in the awareness of an osteointegrating bone screw without remaining foreign material in the body, was excellent.

## Figures and Tables

**Figure 1 jcm-11-01384-f001:**
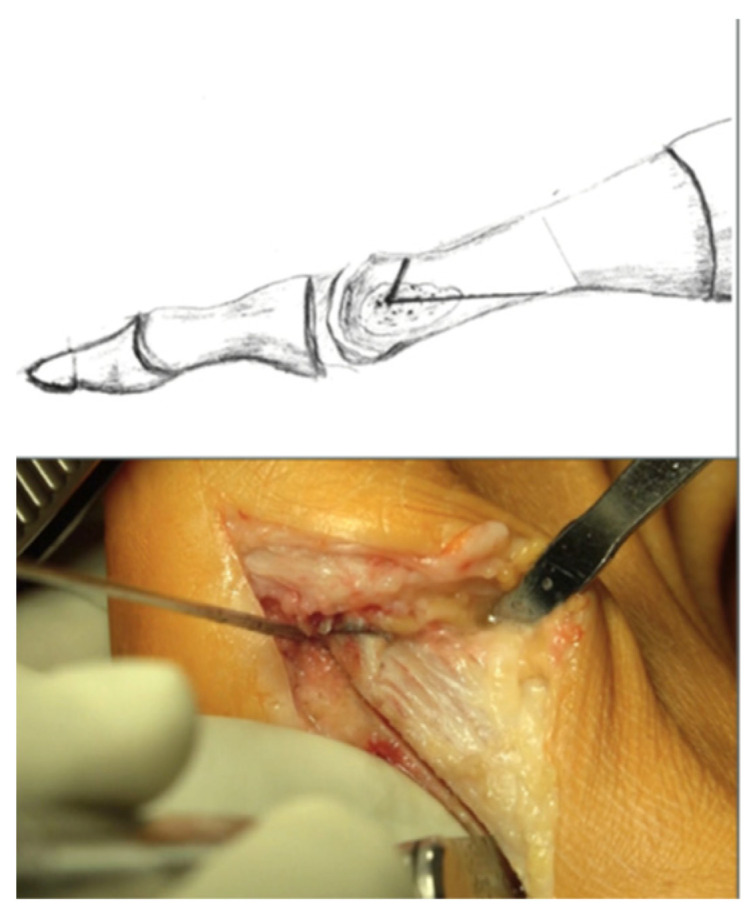
Shows the Reversed L-Shape osteotomy with a long plantar limb.

**Figure 2 jcm-11-01384-f002:**
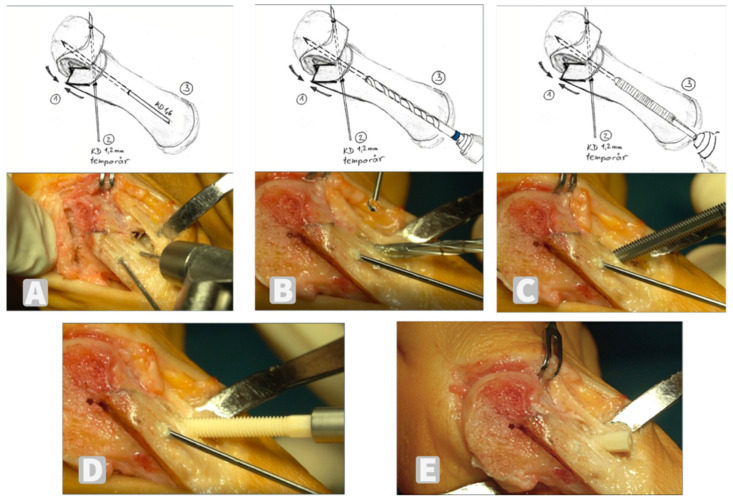
The surgical steps: moving the head fragment laterally ① and temporary stabilization with a 1.2 mm K-wire ②; a 1.6 mm K-wire is inserted centrally in the metatarsal bone ③ from dorsal proximal to plantar distal (**A**); overdrill the core hole step by step (**B**); cut the thread (**C**); insert the Shark Screw^®^ (surgebright GmbH, Lichtenberg, Austria) (**D**); removal of the protruding part of the screw at the level of the bone shaft with the oscillating saw (**E**).

**Figure 3 jcm-11-01384-f003:**
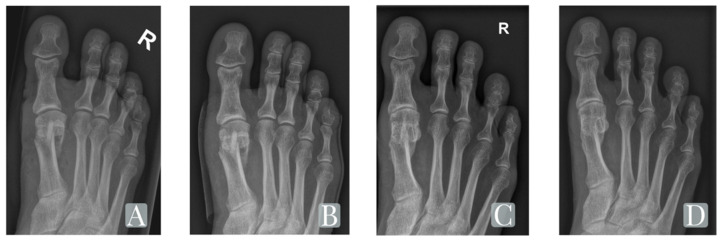
The radiological progression and osteointegration of the Shark Screw^®^ (surgebright GmbH, Lichtenberg, Austria) in a right (R) forefoot postop (**A**), at 6 weeks (**B**), 6 months (**C**), and 16 months (**D**) follow-up.

**Table 1 jcm-11-01384-t001:** Characteristics of patients in this study.

Patient Characteristics	
Number of Patients	n = 15
Age (y)	56.3 (±12.5)
Sex (m/f)	3/12
Follow up (mo)	17 (±8.5)
Lost to FU	n = 1
Previous operations	none
IMA preop	12.6° (±3.2°)
HVA preop	24.8° (±4.9°)

**Table 2 jcm-11-01384-t002:** Inclusion and exclusion criteria for Chevron osteotomy.

Inclusion Criteria	Exclusion Criteria
symptomatic hallux valgus deformity	MTP 1: cartilage damage grade III + IV ROM below 50 degrees hallux valgus angle over 60 degrees
intermetatarsal angle of up to 20 degrees	TMT 1: vertical or horizontal instability ^a^ hypermobility or arthritis
minimum age of 18	peripheral vascular diseases peripheral neuropathy any kind of consuming disease

Abbreviations: MTP1, first metatarsophalangeal joint; TMT1, first tarsometatarsal joint; ^a^ indicated via clinical examination or radiographic signs in the lateral radiograph.

**Table 3 jcm-11-01384-t003:** Correction of IMA and HVA and improvement in pain level and AOFAS score. Comparison baseline with postoperative and 12-month follow-up.

	Baseline	Postop.	12 Months	*p*-Value Baseline vs. Postop./12 mo.
Total Number	15	15	13	
IM-Angle	12.6° (±3.2°)	4.8° (±1.3°)	5.9° (±1.9°)	<0.001/<0.001
HV-Angle	24.8° (±4.9°)	7.2° (±4.4°)	9.9° (±7.0°)	<0.001/<0.001
AOFAS-Score (0–100)	51.6 (±15.2)	61.3 (±13.2)	90.9 (±10.3)	0.037/<0.001
Pain-NRS (0–10)	6.5 (±1.5)	5.4 (±2.1)	1 (±1.4)	0.058/<0.001

Abbreviations: ROM, range of motion degrees); AOFAS, American Orthopedic Foot and Ankle society score (points); IM-angle, intermetatarsal (degrees); HV-angle, Hallux valgus (degrees); NRS, numeric rating scale; paired *t*-test.

## Data Availability

The data presented in this study are available on request from the corresponding author.

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
