# Peer review of "The Application of an Allogenic Bone Screw for Stabilization of a Modified Chevron Osteotomy: A Prospective Analysis"

_jcm, 2022, doi:10.3390/jcm11051384_

Round 1

Reviewer 1 Report

Material and Methods

Authors have to justify the sample size with similar manuscripts written on the previous studies.

 Statistical analysis.- minimal detectable change and standar error of measurement or similar analysis have to be assessed to improve the quality. of the paper.

Discussion

A deep and substantial comparison should be made between bone grades and/or milimeters corrections achieved in the present investigation and those of other authors.

It would be important to add on Limitation section, that the year +-SD of participants is wide; however, sample size is short; and by these reasons, the results have to be interpretable with caution, 

Reviewer 2 Report

Thank you for this helpful contribution. I applaud the effort of promoting these studies.

The manuscript shows interesting results concerning the application of an allogenic bone screw for stabilization of a modified Chevron osteotomy and they deserve to be known by the scientific community. The manuscript is very interesting and I have just a few comments that I believe will improve this paper.

In addition to the below comments, the manuscript would benefit from additional proof reading for grammar and syntax. We recommended read the information on how to format figures, tables and additional files. The author should review the table presentation format. I write some comments below that could benefit the article:

The Introduction section is correct and adequate. The Material & Methods section:  Check table 1 and table 2. It has a format error. There is a different typeface in the tables. Line 124, line space.

The Results section:  Check table 3. It has a format error (Subtitle tables). There is a different typeface in the tables.

The conclusions section must be more concrete. Please, this section should be shortened. Delete line 274-275. I hope that the authors will attend to make the proposed changes to accept the manuscript.

References: The authors should expand journal references. There is no limit to the number of references cited. These are some of the references that should include: *Medium-term outcomes of Chevron osteotomy for hallux valgus correction in a spanish population: Radiologic and clinical parameters and patient satisfaction. J Am Podiatr Med Assoc. 2021 May 1;111(3) doi: 10.7547/18-159. PMID: 33196776

Overall, it is a well-written manuscript with exciting results. With the proposed corrections.

Thank you for this invitation to provide a review to evaluate this article. I am delighted to have been invited to review this work.

Congratulations on this wonderful article!!

Reviewer 3 Report

Dear Authors: I congratulate with You for Your work and for the paper. I recommend minor revision to be done.

In particular, if You know/can investigate, I would like to know which is the cost os the shark screw (to be used in all cases) against the cost of metallic screw removal (to be performed in which percentage of patients? in my practice I am removing screws in less than 3% of patients..).

best regards,
